# Multi-step Retriever-Reader Interaction for Scalable Open-domain Question Answering

**Rajarshi Das[1], Shehzaad Dhuliawala[2], Manzil Zaheer[3] & Andrew McCallum[1]**
{rajarshi,mccallum}@cs.umass.edu
shehzaad.dhuliawala@microsoft.com, manzil@zaheer.ml
[1] University of Massachusetts, Amherst, [2] Microsoft Research, Montréal
[3] Google AI, Mountain View

## Abstract

This paper introduces a new framework for open-domain question answering in which the retriever and the reader *iteratively interact* with each other. The framework is agnostic to the architecture of the machine reading model, only requiring access to the token-level hidden representations of the reader. The retriever uses fast nearest neighbor search to scale to corpora containing millions of paragraphs. A gated recurrent unit updates the query at each step conditioned on the *state* of the reader and the *reformulated* query is used to re-rank the paragraphs by the retriever. We conduct analysis and show that iterative interaction helps in retrieving informative paragraphs from the corpus. Finally, we show that our multi-step-reasoning framework brings consistent improvement when applied to two widely used reader architectures (DR.QA and BIDAF) on various large open-domain datasets — TRIVIAQA-unfiltered, QUASAR-T, SEARCHQA, and SQUAD-open[1].

## 1 Introduction

Open-domain question answering (QA) (Voorhees et al., 1999) involves a *retriever* for selecting relevant context from a large corpora of text (e.g. Wikipedia) and a machine reading comprehension (MRC) model for 'reasoning' on the retrieved context. A lot of effort has been put into designing sophisticated neural MRC architectures for reading short context (e.g. a single paragraph), with much success (Wang & Jiang, 2017; Seo et al., 2017; Xiong et al., 2017; Wang et al., 2018c; Yu et al., 2018, inter alia). However, the performance of such systems degrades significantly when combined with a retriever in open domain settings. For example, the exact match accuracy of DrQA (Chen et al., 2017), on the SQUAD dataset (Rajpurkar et al., 2016) degrades from 69.5% to 28.4% in open-domain settings. The primary reason for this degradation in performance is due to the retriever's failure to find the relevant paragraphs for the machine reading model (Htut et al., 2018).

We propose the following two desiderata for a general purpose open-domain QA system - (a) The retriever model should be *fast*, since it has to find the relevant context from a very large text corpora and give it to the more sophisticated and computationally expensive MRC model (b) Secondly, the retriever and reader models should be interactive, i.e. if the reader model is unable to find the answer from the initial retrieved context, the retriever should be able to *learn* to provide more relevant context to the reader. Open-domain QA systems such as R³ (Wang et al., 2018a) and DS-QA (Lin et al., 2018) have sophisticated retriever models where the reader and retriever are jointly trained. However, their retriever computes *question-dependent* paragraph representation which is then encoded by running an expensive recurrent neural network over the tokens in the paragraph. Since the retriever has to rank a lot of paragraphs, this design does not scale to large corporas. One the other hand, the retriever model of QA systems such as DrQA (Chen et al., 2017) is based on a tf-idf retriever, but they lack trainable parameters and are consequently unable to recover from mistakes.

This paper introduces an open domain architecture in which the retriever and reader iteratively interact with each other. Our model first pre-computes and caches representation of context (paragraph). These representations are independent of the query unlike recent architectures (Wang et al., 2018a;

---

[1]Code and pretrained models are available at https://github.com/rajarshd/Multi-Step-Reasoning

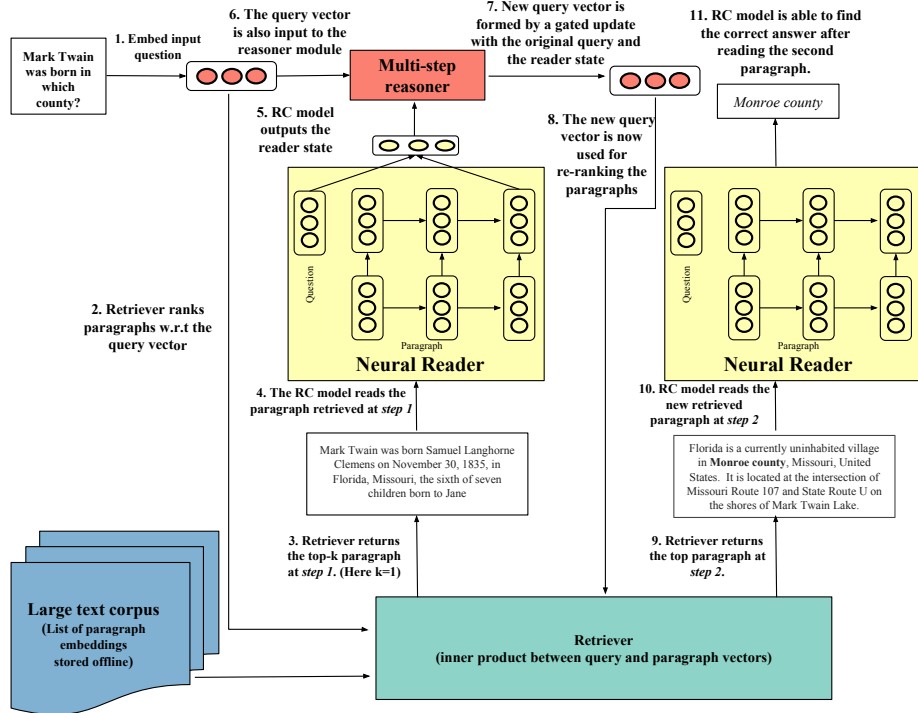

Figure 1: Our framework unrolled for two steps. The initial query is encoded and the retriever sends the top-*k* paragraphs to the reader. The *multi-step-reasoner* component of our model takes in the internal state of the reader model and the previous query vector and does a gated update to produce a *reformulated* query. This new query vector is used by the retriever to re-rank the paragraphs and send different paragraphs to the reader. Thus the multi-step-reasoner facilitates *iterative* interaction between the retriever (search engine) and the reader (QA model)

Lin et al., 2018) and hence can be computed and stored offline. Given an input question, the retriever performs *fast* inner product search to find the most relevant contexts. The highest ranked contexts are then passed to the neural MRC model. Our architecture is agnostic to the choice of the reader architecture and we show that multi-step-reasoning increases performance of two state-of-the-art MRC architectures - DrQA (Chen et al., 2017) and BiDAF (Seo et al., 2017).

It is possible that the answer might not exist in the initial retrieved paragraphs or that the model would need to combine information across multiple paragraphs (Wang et al., 2018b). We equip the reader with an additional gated recurrent unit (Cho et al., 2014) which takes in the state of the reader and the current query vector and generates a new query vector. This new query vector is then used by the retriever model to re-rank the context. This allows the model to read new paragraphs and combine evidence across multiple paragraphs. Since the retriever makes a 'hard selection' of paragraphs to send to the reader, we train the retriever and the reader jointly using reinforcement learning.

Our architecture draws inspiration from how students are instructed to take reading comprehension tests (Cunningham & Shablak, 1975; Bishop et al., 2006; Duggan & Payne, 2009). Given a document containing multiple paragraphs and a set of questions which can be answered from the document, (a) the student quickly skims the paragraphs, (b) then for each question, she finds the most relevant paragraphs that she thinks will answer the question. (c) She then carefully reads the chosen paragraph to answer the question (d) However, if the chosen paragraph does not answer the question, then given the question and the knowledge of what she has read till now, she decides which paragraph to read next. Step (a) is akin to our model encoding and storing the question independent paragraph representations and step (b) corresponds to the inner product search to find the relevant context. The reading of the context by the sophisticated neural machine reader corresponds to step (c) and the last step corresponds to the iterative (multi-step) interaction between the retriever and the reader.

To summarize, this paper makes the following contributions: (a) We introduce a new framework for open-domain QA in which the retriever and reader *iteratively* interact with each other via a novel multi-step-reasoning component allowing it to retrieve and combine information from multiple paragraphs.

(b) Our paragraph representations are independent of the query which makes our architecture highly scalable and we empirically demonstrate it by running large scale experiments over millions of paragraphs. (c) Lastly, our framework is agnostic to the architecture of the reader and we show improvements on two widely used neural reading comprehension models.

## 2 BASIC COMPONENTS OF OUR MODEL

The architecture of our model consists of three main components - (a) paragraph retriever - that computes a relevance score for each paragraph w.r.t a given query and ranks them according to the computed score. (b) reader - a more sophisticated neural machine reading model that receives few top-ranked paragraphs from the retriever and outputs a span of text as a possible answer to the query and (c) multi-step-reasoner - a gated recurrent unit that facilitates iterative interaction between the retriever and the reader.

Formally, the input to our model is a natural language question $Q = q_1, q_2, \ldots, q_n$ consisting of $n$ tokens and a set of paragraphs $P = \{p_1, p_2, \ldots p_K\}$. Our model extracts a span of text $a$ as the answer to the question from the paragraphs in P. Note that the set of paragraphs in P can be the paragraphs in a set of documents retrieved by a search engine or it could be *all* the paragraphs in a large text corpus such as Wikipedia. Next we describe each individual component of our model.

### 2.1 PARAGRAPH RETRIEVER

The paragraph retriever computes a score for how likely a paragraph is to contain an answer to a given question. The paragraph representations are computed independent of the query and once computed, they are not updated. This allows us to cache the representations and store them offline. The relevance score of a paragraph is computed as a inner product between the paragraph and the query vectors. The paragraph and query representations are computed as follows.

Given a paragraph $p = \{p_1, p_2, \ldots, p_m\}$ consisting of $m$ tokens, a multi-layer recurrent neural network encodes each tokens in the paragraph — $\{\mathbf{p}_1, \mathbf{p}_2, \ldots, \mathbf{p}_m\} = \text{RNN}(\{p_1, p_2, \ldots, p_m\})$, where $\mathbf{p}_j \in \mathbb{R}^{2d}$ encodes useful contextual information around the $j$-th token. Specifically we choose to use a multi-layer bidirectional long-short term memory network (LSTM) (Hochreiter & Schmidhuber, 1997) and take $\mathbf{p}_j$ as the hidden units in the last layer of the RNN. We concatenate the representation computed by the forward and the backward LSTM. To compute a single paragraph vector $\mathbf{p} \in \mathbb{R}^{2d}$ from all the token representations, we combine them using weights $b_j \in \mathbb{R}$

$$b_j = \frac{\exp(\mathbf{w} \cdot \mathbf{p_j})}{\sum_{j'} \exp(\mathbf{w} \cdot \mathbf{p}_{j'})} \qquad\qquad \mathbf{p} = W_s \sum_{j'} b_{j'} \cdot \mathbf{p_{j'}}$$

Here $b_j$ encodes the importance of each token and $\mathbf{w} \in \mathbb{R}^{2d}, \mathbf{W_s} \in \mathbb{R}^{2d \times 2d}$ are learned weights. The query $q = \{q_1, q_2, \ldots, q_n\}$ is encoded by another network with the same architecture to obtain a query vector $\mathbf{q} \in \mathbb{R}^{2d}$. Next the relevance score of a paragraph w.r.t the query $(\text{score}(\mathbf{p}, \mathbf{q}) \in \mathbb{R})$ is computed by a simple inner product — $\text{score}(\mathbf{p}, \mathbf{q}) = \langle \mathbf{p}, \mathbf{q} \rangle$. The paragraph retriever then returns the top scoring $k$ paragraphs to the reader.

**Fast inner product search.** Inner product can be efficiently computed on a GPU, however in our experiments with large corpus with over million paragraphs, (a) the paragraph vectors require more memory than available in a single commodity GPU (b) it is computationally wasteful to compute inner products with all paragraph vectors and we can do better by leveraging fast nearest neighbor (NN) search algorithms. Since the retriever has to find the $k$ paragraphs with the highest inner products w.r.t query, our problem essentially reduces to maximum inner product search (MIPS). There exists data structures for fast NN search in metric spaces such as Cover Trees (Beygelzimer et al., 2006). But we cannot use them directly, since triangle equality does not hold in inner product space. Borrowing ideas from Bachrach et al. (2014); Zaheer et al. (2019), we propose to use nearest neighbor (NN) search algorithms to perform the MIPS in sublinear time as follows.

Let $\mathbf{u}$ be an upper bound for the L2 norm for all paragraphs, i.e. $\mathbf{u} \geq \|\mathbf{p}\|, \forall \mathbf{p}$. Next, we modify the original paragraph and query vectors such that searching for the k-nearest neighbor w.r.t L2 distance with the modified vectors is equivalent to finding the $k$ nearest (original) paragraph vectors in the inner-product space.

Define the augmented paragraph vectors as $\tilde{\mathbf{p}}_{\mathbf{i}} = [\mathbf{p}_{\mathbf{i}}; \sqrt{\mathbf{u}^2 - \|\mathbf{p}_{\mathbf{i}}\|^2}]$ and augmented query vector as $\tilde{\mathbf{q}} = [\mathbf{q}; 0]$, where $[;]$ denotes concatenation. Note, with this transformation, $\langle \tilde{\mathbf{p}}_{\mathbf{i}}, \tilde{\mathbf{q}} \rangle = \langle \mathbf{p}_{\mathbf{i}}, \mathbf{q} \rangle$. Now,

$$
\begin{aligned}
\text{NN}(\tilde{\mathbf{p}}_{\mathbf{i}}, \tilde{\mathbf{q}}) &= \arg\min_i \|\tilde{\mathbf{p}}_{\mathbf{i}} - \tilde{\mathbf{q}}\|^2 \\
&= \arg\min_i \|\tilde{\mathbf{p}}_{\mathbf{i}}\|^2 + \|\tilde{\mathbf{q}}\|^2 - 2\langle \tilde{\mathbf{p}}_{\mathbf{i}}, \tilde{\mathbf{q}} \rangle \\
&= \arg\min_i (\mathbf{u}^2 - 2\langle \tilde{\mathbf{p}}_{\mathbf{i}}, \tilde{\mathbf{q}} \rangle) \\
&= \arg\max_i \langle \tilde{\mathbf{p}}_{\mathbf{i}}, \tilde{\mathbf{q}} \rangle = \arg\max_i \langle \mathbf{p}_{\mathbf{i}}, \mathbf{q} \rangle = \text{MIPS}(\mathbf{p}_{\mathbf{i}}, \mathbf{q})
\end{aligned}
$$

Many exact NN search algorithms can find k-NN in time (sublinear) logarithmic in number of paragraphs (Beygelzimer et al., 2006; Zaheer et al., 2019), after an one-time preprocessing. In our architecture, the preprocessing can be done because the set of all paragraphs is fixed for all the queries. We chose to use SGTree (Zaheer et al., 2019) to perform the NN/MIPS due to its fast construction time and competitive performance reducing the search to logarithmic time per query.

Note that our paragraph representations are independent of the given query and are *never updated* once they are trained. After training is completed, we cache the $\mathbf{p} \in \mathbb{R}^{2d}$ vectors. This is unlike many recent work in open-domain QA. For example, in R$^3$ (Wang et al., 2018a), the retriever uses Match-LSTM model (Wang & Jiang, 2017) to compute question-matching representations for each token in the passage. The paragraph representations are then obtained by running a bi-directional RNN over these matched representation. Although powerful, this architecture design will not scale in open-domain settings where the retriever has to re-compute new representations for possibly millions of paragraphs for every new question. In contrast, once training is over, we cache the paragraph representation and use it for every query at test time.

**Training** - Following previous work (Htut et al., 2018; Lin et al., 2018; Wang et al., 2018a), we gather labels for paragraphs during training using distant supervision (Mintz et al., 2009). A paragraph that contains the exact ground truth answer string is labeled as an positive example. For a positive (negative) labeled paragraph, we maximize (minimize) the $\log(\sigma(\text{score}(p, q)))$. The number of layers of the bi-directional LSTM encoder is set to three and we use Adam (Kingma & Ba, 2014) for optimization. Once training is done, we pre-compute and cache the paragraph representation of each dataset in our experiments.

## 2.2 MACHINE READER

The reader is a sophisticated neural machine reading comprehension (MRC) model that takes in the top few paragraphs sent by the retriever and outputs a span of answer text. Our model is agnostic to the exact architecture of the reader and we perform experiments to show the efficacy of our model on two state-of-the-art neural machine reading models - DrQA and BiDAF.

MRC models designed for SQUAD (Rajpurkar et al., 2016), NewsQA (Trischler et al., 2016), etc operate on a single paragraph. However it has been shown that reading and aggregating evidence across multiple paragraphs is important when doing QA over larger evidence set (e.g. full Wikipedia document) (Swayamdipta et al., 2018; Clark & Gardner, 2018) and in open domain settings (Wang et al., 2018b). Most MRC models compute a start and end scores for each token in the paragraph that represents how likely a token is the start/end of an answer span. To gather evidence across multiple paragraphs sent by the retriever, we normalize the start/end scores across the paragraphs. Furthermore a text span can appear multiple times in a paragraph. To give importance to all answer spans in the text, our objective aggregates (sums) the log-probability of the score for each answer position. Let $\mathcal{I}(w, p)$ denote the token start positions where the answer span appears in the paragraph $p$ and let $w_S$ be the starting word of the answer span. Our model maximizes the sum of the following objective for the start and end word of an answer spans as follows. (For brevity, we only show the objective for the starting word ($w_s$) of the span.)

$$
\log \left( \frac{\sum_{j \in \mathcal{P}} \sum_{k \in \mathcal{I}(w_s, p_j)} \exp(\text{score}_{\text{start}}(k, j))}{\sum_{j \in \mathcal{P}} \sum_{i=1}^{n_j} \exp(\text{score}_{\text{start}}(i, j))} \right)
$$

Here, $\mathcal{P}$ denotes the set of all top-ranked paragraphs by the retriever, $n_j$ denotes the number of tokens in paragraph $j$ and $\text{score}_{\text{start}}(k, j)$ denotes the start score of the $k$-th token in the $j$-th paragraph.

**Score aggregation during inference.** During inference, following Chen et al. (2017); Seo et al. (2017), the score of a span that starts at token position $i$ and ends at position $j$ of paragraph $p$ is given by the sum of the $\text{score}_{\text{start}}(i, p)$ and $\text{score}_{\text{end}}(j, p)$. During inference, we score spans up to a pre-defined maximum length of 15. Further, we add the scores of spans across different paragraphs (even if they are retrieved at different steps), if they have the same surface form. That is, if the same span (e.g. "Barack Obama") occurs multiple times in several paragraphs, we sum the individual span scores. However, to ensure tractability, we only consider the top 10 scoring spans in each paragraph for aggregation. It should be noted that the changes we made to aggregate evidence over multiple paragraphs and mentions of spans needs *no change* to the original MRC architecture.

## 3 MULTI-STEP-REASONER

A novel component of our open-domain architecture is the multi-step-reasoner which facilitates *iterative interaction* between the retriever and the machine reader. The multi-step-reasoner a module comprised of a gated recurrent unit (Cho et al., 2014) which takes in the current state of the reader and the current query vector and does a gated update to produce a reformulated query. The reformulated query is sent back to the retriever which uses it to re-rank the paragraphs in the corpus. Since the gated update of the multi-step-reasoner, conditions on the current state of the machine reader, this multi-step interaction provides a way for the search engine (retriever) and the QA model (reader) to communicate with each other. This can also be seen as an instance of two agents cooperating via communication to solve a task (Lazaridou et al., 2017; Lee et al., 2018; Cao et al., 2018).

More formally, let $\mathbf{q_t} \in \mathbb{R}^{2d}$ be the current query representation which was most recently used by the paragraph *retriever* to score the paragraphs in the corpus. The multi-step-reasoner also has access to the *reader* state which is computed from the hidden memory vectors of the reader. The reader state captures the current information that the reader has encoded after reading the paragraphs that was sent by the retriever. Next we show how the reader state is computed.

Span extractive machine reading architectures compute a hidden representation for each token in the paragraph. Our framework needs access to these hidden representations to compute the reader state.

Let $\mathbf{m_j} \in \mathbb{R}^{2p}$ be the hidden vector associated with the $j$-th token in the paragraph. Let $\mathbf{L} \in \mathbb{R}^{2p}$ be the final query representation of the reader model. $\mathbf{L}$ is usually created by some pooling operation on the hidden representation of each question token. The reader state $\mathbf{S} \in \mathbb{R}^{2p}$ is computed from each of the hidden vectors $\mathbf{m_j}$ and $\mathbf{L}$ by first computing soft-attention weights between each paragraph token, followed by combining each $\mathbf{m_j}$ with the soft attention weights.

$$\alpha_{\mathbf{j}} = \frac{\exp\left(\mathbf{m_j} \cdot \mathbf{L}\right)}{\sum_{j'} \exp\left(\mathbf{m_j'} \cdot \mathbf{L}\right)} \qquad\qquad \mathbf{S} = \sum_j \left(\alpha_{\mathbf{j}} \cdot \mathbf{m_j}\right)$$

Finally, the new reformulated query $\mathbf{q_{t+1}} \in \mathbb{R}^{2d}$ for the paragraph retriever is calculated by the multi-step-reasoner module as follows —

$$\mathbf{q}_{\mathbf{t+1}}' = \text{GRU}\left(\mathbf{q_t}, \mathbf{S}\right)$$
$$\mathbf{q_{t+1}} = \text{FFN}(\mathbf{q}_{\mathbf{t+1}}')$$

In our architecture, we used a 3 layer GRU network, followed by a one layer feed forward network (FFN) with a ReLU non-linearity. The gated update ensures that relevant information from $\mathbf{q_t}$ is preserved and new and useful information from the reader state $\mathbf{S}$ is added to the reformulated query.

### 3.1 TRAINING

There exists no supervision for training the query reformulation of the multi-step-reasoner. Instead, we employ *reinforcement learning* (RL) and train it by how well the reader performs after reading the new set of paragraphs retrieved by the modified query. We define the problem as a deterministic finite horizon Partially Observed Markov decision process (POMDP). The components of POMDP are —

**States**. A state in the state space consists of the entire text corpora, the query, the answer, and $k$ selected paragraphs. The reader is part of the environment and is fully described given the current state, i.e. selected paragraphs and the query.

**Observations**. The agent only observes a function of the current state. In particular, to the agent only the query vector and the memory of the reader model is shown, which are a function of the current state. Intuitively, this represents the information encoded by the machine reader model after reading the top $k$ paragraphs sent by the retriever in current step.

**Actions**. The set of *all* paragraphs in the text corpora forms the action space. The retriever scores all paragraphs w.r.t the current query and selects the top $k$ paragraphs to send to the reader model. We treat $k$ as a hyper-parameter in our experiments.

**Reward**. At every step, the reward is measured by how well the answer extracted by the reader model matches to the ground-truth answer. We use the $F_1$ score (calculated by word overlap between prediction and ground-truth) as the reward at each step.

**Transition**. The environment evolves deterministically after reading the paragraphs sent by the retriever.

Our policy $\pi$ is parameterized by the GRU and the FFN of the multi-step-reasoner. Let $r_t$ denote the reward ($F_1$ score) returned by the environment at time $t$. We directly optimize the parameters to maximize the expected reward given by — $J(\theta) = \mathbb{E}_\pi \left[ \sum_{t=1}^{T} r_t \right]$. Here $T$ denotes the number of steps of interactions between the retriever and the reader. We treat the reward at each step equally and do not apply any discounting. Motivated by the REINFORCE (Williams, 1992) algorithm, we compute the gradient of our objective as

$$\nabla_\theta J(\theta) = \mathbb{E}_\pi \left[ \sum_{t=1}^{T} r_t \cdot \log(\pi_\theta(p_t \mid q)) \right]$$

$$\pi_\theta(p_t \mid q) = \mathrm{softmax}(\mathrm{score}(p_t, q))$$

Here, $p_t$ is the top-ranked paragraph returned by the retriever at time $t$ and the probability that the current policy assigns to $p_t$ is computed by normalizing the scores with a softmax. It is usual practice to add a variance reduction baseline (e.g. average of rewards in the minibatch) for stable training, but we found this significantly degrades the final performance. We think this is because, in QA, a minibatch consists of questions of varying difficulty and hence the rewards in a batch itself have high variance. This is similar to findings by Shen et al. (2017) for closed domain-QA (e.g. SQUAD).

**Pretraining the multi-step-reasoner**. We also find that pre-training the multi-step-reasoner before fine tuning with RL to be effective. We train to rank the similarity between the reformulated query vector and one of the distantly-supervised *correct* paragraph vectors higher than the score of a randomly sampled paragraph vector. Specifically, let $\mathbf{q}$ denote the query vector, $\mathbf{p}^*, \mathbf{p}'$ denote the paragraph representation of paragraphs containing the correct answer string and a random sampled paragraph respectively. We follow the Bayesian Personalized Ranking approach of Rendle et al. (2009) and maximize $\log(\sigma(\mathbf{q}^\top \mathbf{p}^* - \mathbf{q}^\top \mathbf{p}'))$. The paragraph representations $\mathbf{p}^*, \mathbf{p}'$ are kept fixed and gradient only flows through $\mathbf{q}$ to the parameters of the multi-step-reasoner.

## 3.2 PUTTING IT ALL TOGETHER

Our open-domain architecture is summarized above in Algorithm 1. Given a large text corpora (as a list of paragraphs), the corresponding paragraph embeddings (which can be trained by the procedure in (§ 2.1)) and hyper-parameters $(T, k)$, our model $\mathcal{M}$ returns a text span $a$ as answer. The multi-step interaction between the retriever and reader can be best understood by the `for` loop in line 2 of algorithm 1. The initial query $\mathbf{q_0}$ is first used to rank *all* the paragraphs in the corpus (line 3), followed by which the top $k$ paragraphs are sent to the reader (line 4). The reader returns the answer span (with an associated score for the span) and also its internal state (line 5). The GRU network then takes in the current query and the reader state to produce the updated query which is then passed to the retriever (line 6). The retriever uses this updated query to again re-rank the paragraphs and the entire process is repeated for T steps. At the end of T steps, the model returns the span with the highest score returned by the reader model. The reader is trained using supervised learning (using the correct spans as supervision) and the parameters of the GRU network are trained using reinforcement learning. During training, we first pre-train the reader model by setting the number of multi-step reasoning steps (T = 1). After the training converges, we freeze the parameters of the reader model and train the parameters of the GRU network using policy gradients. The output of the reader model is used to generate the reward which is used to train the policy network.

---

**Algorithm 1** Multi-step reasoning for open-domain QA

---

**Input:** Question text $Q_{text}$, Text corpus (as a list of paragraphs) $\mathcal{T}$, Paragraph embeddings $\mathcal{P}$ (list of paragraph embeddings for each paragraph in $\mathcal{T}$), Model $\mathcal{M}$, Number of multi-steps T, number of top-ranked paragraphs $k$
**Output:** Answer span $a$

1: $\mathbf{q}_0 \leftarrow$ `encode_query`$(Q_{text})$      # (§ 2.1)
2: **for** t in `range`(T) **do**
3:    $\{p_1, p_2, \ldots p_k\} \leftarrow \mathcal{M}$.`retriever.score_paras`$(\mathbf{q_t}, \mathcal{P}, k)$    # each $p_i$ denotes a paragraph id
4:    $P_{text}^{1 \ldots k} \leftarrow$ `get_text`$(\mathcal{T}, \{p_1, p_2, \ldots p_k\})$    # get text for the top paragraphs
5:    $a_t, s_t, \mathbf{S_t} \leftarrow \mathcal{M}$.`reader.read`$(P_{text}^{1 \ldots k}, Q_{text})$    # (§ 2.2); $\mathbf{S_t}$ denotes reader state
                                           # $a_t$ denotes answer span
                                           # $s_t$ denotes the score of the span
6:    $\mathbf{q_t} \leftarrow \mathcal{M}$.`multi_step_reasoner.GRU`$(\mathbf{q_t}, \mathbf{S_t})$
                                           # (§ 3); Query reformulation step
7: **end for**
8: **return** answer span $a$ with highest score

---

# 4 RELATED WORK

**Open domain QA** is a well-established task that dates back to few decades of reasearch. For example the BASEBALL system (Green Jr et al., 1961) aimed at answering open domain question albeit for a specific domain. Open-domain QA has been popularized by the Trec-8 task (Voorhees et al., 1999) and has recently gained significant traction due to the introduction of various datasets (Dunn et al., 2017; Dhingra et al., 2017; Joshi et al., 2017). In many open-domain systems (Chen et al., 2017), the retriever is a simple IR based system (e.g. tfidf retriever) with no trainable parameters and hence the retriever cannot overcome from its mistakes. Recent work such as $R^3$ (Wang et al., 2018a), DS-QA (Lin et al., 2018) use a trained retriever and have shown improvement in performance. However they form query dependent paragraph representation and such architectures will not scale to full open-domain settings where the retriever has to rank millions of paragraphs. and neither do they support iterative reasoning thereby failing to recover from any mistakes made by the ranker or where evidence needs to be aggregated across multiple paragraphs.

**Query Reformulation** by augmenting the original query with terms from the top-$k$ retrieved document (Xu & Croft, 1996; Lavrenko & Croft, 2001) has proven to be very effective in information retrieval. Instead of using such automatic relevance feedback, Nogueira & Cho (2017) train a query reformulation model to maximize the recall of a IR system using RL and Pfeiffer et al. (2018) showed that query refinement is effective for IR in bio-medical domain. The main difference between this work and ours is that our model directly optimizes to improve the performance of a question answering system. However, as we show, we still see an improvement in the recall of our retriever (§ 5.1). Perhaps the most related to our work is Active Question Answering (AQA) (Buck et al., 2018), which use reformulation of the natural language query to improve the performance of a BiDAF reader model on SEARCHQA. The main difference between AQA is that we reformulate the query in vector space. We compare with AQA and show that our model achieves significantly better performance.

**Iterative Reasoning** has shown significant improvements in models using memory networks (Sukhbaatar et al., 2015; Miller et al., 2016) for question answering in text and knowledge bases (Das et al., 2017; Yang et al., 2017; Das et al., 2018). Our model can be viewed as a type of controller update step of memory network type of inference. Recently, iterative reasoning has shown to be effective in reading comprehension in single paragraph setting, where the model reads the same paragraph iteratively (Shen et al., 2017; Liu et al., 2017). Our work can be seen as a strict generalization of these in a more realistic, open-domain setting.

**Nearest Neighbor Search** - Computing fast nearest neighbor (NN) search is a fundamental requirement in many applications. We used exact k-NN search using SGTree (Zaheer et al., 2019) which has been shown to perform better than other exact k-NN strategies such as Cover Tree (Beygelzimer et al., 2006), P-DCI (Li & Malik, 2017) and other approximate NN search techniques such as RP-Tree (Dasgupta & Sinha, 2013), HNSW (Malkov & Yashunin, 2018). In general, our proposed multi-step-reasoner framework is not coupled to any particular kind of k-NN technique.

| Model | Quasar-T | | SearchQA | | TRIVIAQA-unfiltered | | SQUAD-open | |
|---|---|---|---|---|---|---|---|---|
| | EM | F1 | EM | F1 | EM | F1 | EM | F1 |
| GA (Dhingra et al., 2016) | 26.4 | 26.4 | - | - | - | - | - | - |
| BIDAF (Seo et al., 2017) | 25.9 | 28.5 | 28.6 | 34.6 | - | - | - | - |
| AQA (Buck et al., 2018) | - | - | 40.5 | 47.4 | - | - | - | - |
| R³ (Wang et al., 2018a) | 35.3 | 41.7 | 49.0 | 55.3 | 47.3 | 53.7 | 29.1 | 37.5 |
| DS-QA* (Lin et al., 2018) | 37.27 | 43.63 | **58.5** | **64.5** | 48.7 | 56.3 | 28.7 | 36.6 |
| MINIMAL (Min et al., 2018) | - | - | - | - | - | - | **34.7** | **42.5** |
| Dr.QA baseline | 36.87 | 45.49 | 51.36 | 58.24 | 48.00 | 52.13 | 27.1 | - |
| multi-step-reasoner (Dr.QA) | **39.53** | **46.67** | 55.01 | 61.61 | 55.93 | 61.66 | 31.93 | 39.22 |
| multi-step-reasoner (BiDAF) | **40.63** | **46.97** | 56.26 | 61.36 | 55.91 | 61.65 | - | - |
| DocumentQA** (Clark & Gardner, 2018) | - | - | - | - | 61.56 | 68.03 | - | - |

* Despite our best efforts, we could not reproduce the results of Ds-QA using their code and hyperparameter settings for Quasar-T.

** The results on the test set of TRIVIAQA-unfiltered were not reported in the original paper. Results obtained from authors via e-mail.

Table 2: Performance on test sets for various datasets

| Model | P@1 | P@3 | P@5 |
|---|---|---|---|
| R³ (Wang et al., 2018a) | 40.3 | 51.3 | 54.5 |
| Our Retriever (initial) | 35.7 | 49.6 | 56.3 |
| + multi-step (7 steps) | **42.9** | **55.5** | **59.3** |

Table 3: Retrieval performance on QUASAR-T. The match-LSTM based retriever of R³ is a more powerful model than our intial retrieval model. However, after few steps of multi-step-reasoner, the performance increases suggesting that re-ranking via query-reformulation is retrieving relevant evidence from the corpus. We report the P@*k* on the last step.

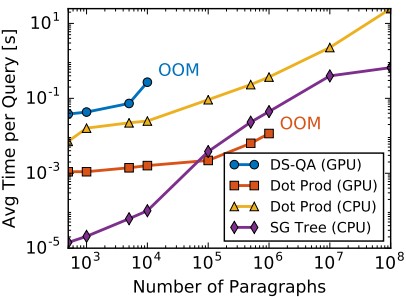

Figure 2: Scalability of retriever.

# 5 EXPERIMENTS

We now present experiments to show the effectiveness of each component of our framework. We experiment on the following large open-domain QA datasets — (a) TRIVIAQA-unfiltered– a version of TRIVIAQA (Joshi et al., 2017) built for open-domain QA. It has much more number of paragraphs than the web/wiki setting of TRIVIAQA. Moreover, there is no guarantee that every document in the evidence will contain the answer making this setting more challenging. (b) TRIVIAQA-open– To test our framework for large scale setting, we combine *all* evidence for every question in the development set. This resulted in a corpus containing **1.68M** paragraphs per question. (c)

| Datasets | #q(test) | #p/q(test) |
|---|---|---|
| SEARCHQA | 27,247 | 49.6 |
| QUASAR-T | 3,000 | 99.8 |
| SQUAD-open | 10,570 | 115.6 |
| TRIVIAQA-unfiltered | 10,790 | 149 |
| TRIVIAQA-open | 11,274 | **1,684,193** |

Table 1: Statistics of various dataset. The second column shows the number of paragraphs for each query.

SEARCHQA (Dunn et al., 2017) – is another open-domain dataset which consists of question-answer pairs crawled from the J! archive. The paragraphs are obtained from 50 web snippets retrieved using the Google search API. (d) QUASAR-T (Dhingra et al., 2017) – consists of 43K open-domain questions where the paragraphs are obtained from the ClueWeb data source. (e) SQUAD-open– We also experimented on the open domain version of the SQUAD dataset. For fair comparison to baselines, our evidence corpus was created by retrieving the top-5 wikipedia documents as returned by the pipeline of Chen et al. (2017). The datasets are summarized in table 1.

## 5.1 PERFORMANCE OF PARAGRAPH RETRIEVER

We first investigate the performance of our paragraph retriever model (§2.1). Our retriever is based on inner product between the query and pre-computed paragraph vectors. This architecture, although more scalable, is less powerful than the retrievers of R³ and DS-QA which compute query-dependent passage representation via soft alignment (attention) before encoding with a bi-LSTM. Results in

table 3 indeed show that retriever of $R^3$ has better P@$k$ than our retriever. We also measure the performance of our retriever after few steps (#steps = 7) of interaction between the retriever and reader. As we can see from table 3, the query reformulation has resulted in better re-ranking and hence an overall improved performance of the retriever.

To test for scalability in open-domain settings, we conduct a synthetic experiment. We plot the wall-clock time (in seconds) taken by the retriever to score and rank paragraphs with a query. To test for scalability, we increase the number of paragraphs ranging from 500 to 100 million and test on a single Titan-X GPU. For our baseline, we use the GPU implementation of the retriever of DS-QA (Lin et al., 2018). For our model, we test on three variants — (a) inner product on GPU, (b) inner-product on CPU and (c) inner-product using SG-Tree. Figure 2 shows the results. Our retriever model on GPU is faster than DS-QA, since the latter perform much more computation compared to just inner-product in our case. Moreover, DS-QA quickly uses up all the available memory and throws a memory error (OOM) by just 100K paragraphs. The inner product operation scales up to 1M paragraph before going OOM. SG-Tree shows very impressive performance even though it operates on CPU and consistently outperforms dot product operation on the CPU.

## 5.2 EFFECTIVENESS OF MULTI-STEP-REASONER

Next we investigate the improvements in QA performance due to multi-step-reasoner. We report the commonly used exact-match (EM) and F1 score on each dataset.

**Performance on benchmark datasets.** Table 2 compares the performance of our model with various competitive baselines on *four* open-domain QA datasets. One of the main observation is that combining multi-step-reasoner with the base Dr.QA reader (Chen et al., 2017) always leads to improved performance. We also perform competitively to most baselines across all datasets.

Figure 3 shows the relative improvements our models achieve on QUASAR-T, SEARCHQA and TRIVIAQA-unfiltered with varying number of steps of interaction between the retriever and the reader. The key takeaway of this experiment is that multiple steps of interaction uniformly increases performance over base model (with no interaction). Different datasets have varying level of difficulties, however, the performance reaches its peak around 5 to 7 steps and does not provide much benefit on further increasing the number of steps.

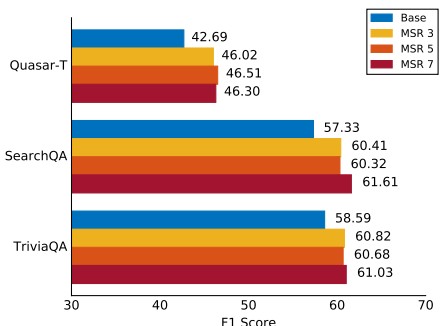

Figure 3: F1 score w.r.t number of steps.

**Large scale experiment on TRIVIAQA-open**. The benchmark datasets for open-domain QA have on an average hundred paragraphs for each question (Table 1). In real open-domain settings, the size of the evidence corpus might be much bigger. To test our model for the same, we created the TRIVIAQA-open setting, in which for each query we combined all the paragraphs in the development set, resulting in 1.6M paragraphs per query. We want to emphasize that the baseline models such as $R^3$ and DS-QA will not scale to this large setting. The baseline DrQA. model with no interaction between the retriever and reader get a score of EM = 37.45 and F1 = 42.16. The same model with 3 steps of interaction achieves a score of EM = **39.76** and F1 = **44.30**. The key takeaways from this experiment are — (a) our framework can scale to settings containing millions of paragraphs, (b) iterative interaction still increase performance even in large scale settings, (c) the overall performance has significantly decreased (from 61 to 44.3 F1), suggesting that large context in open-domain QA is very challenging and should be an active area of research with significant scope for improvement.

## 5.3 ANALYSIS OF RESULTS

This section does further analysis of the results of the model. Specifically we are interested in analyzing if our method is able to gather more relevant evidence as we increase the number of steps of interaction between the retriever and the reader. We conduct this analysis on the development set of SEARCHQA (containing 13,393 queries) and to simplify analysis, we make the retriever choose the top scored single paragraph to send to the reader, at each step. The results are noted in table 4. As we can see from row 1 of table 4, on increasing steps of interaction, the quality of retrieved paragraphs

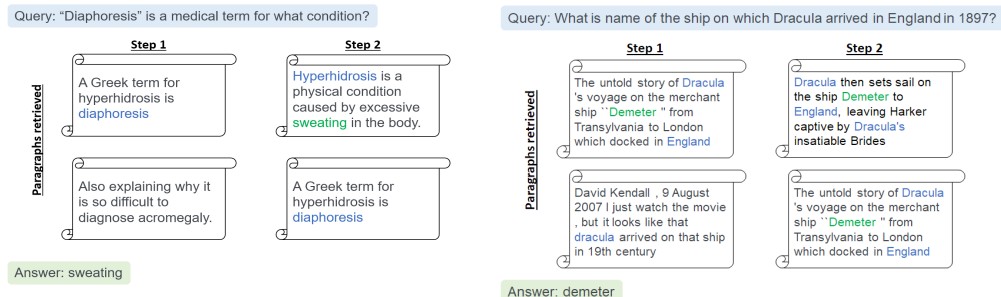

Figure 4: Examples of how multi-step-reasoner iteratively modifies the query by reading context to find more relevant paragraphs. Figure (left) shows an example where the initial retrieved context did not have the answer but the context provided enough hint to get more relevant paragraph in the next step. In figure (right), both the retrieved paragraph have the answer string leading to a boost in the score of the answer span because of score aggregation of spans (§2.2).

| | # steps of interaction | | |
|---|---|---|---|
| | 3 | 5 | 7 |
| # queries where initial retrieved para was incorrect but correct para was retrieved later | 1047 | 1199 | 1270 |
| # queries where correct para wasn't retrieved at all | 3783 | 3594 | 3505 |
| # queries where initial answer was incorrect but recovered later | 490 | 612 | 586 |
| Avg. number of unique paragraphs read across all steps | 1.99 | 2.38 | 3.65 |

Table 4: Analysis of results on the dev. set of SEARCHQA as we increase the number of steps of interaction between the retriever and reader. The retriever at each step sends top-1 paragraph to the reader. A paragraph is correct if it contains the correct answer string.

becomes better, i.e. even though the correct paragraph (containing the answer string) was not retrieved in the first step, the retriever is able to fetch relevant paragraphs later. It is also unsurprising to see, that when correct paragraphs are retrieved, the performance of the reader also increases. To check if the our policy was exploring and retrieving different paragraphs, we counted the mean number of unique paragraphs retrieved across all steps and found that it is high (around 2) when #steps of interaction is small as 3, suggesting that the policy chooses to explore initially. As the number of steps is increased, it increases at slower rate suggesting that if the policy has already found good paragraphs for the reader, it chooses to exploit them rather than choosing to explore.

Figure 4 shows two instances where iterative interaction is helpful. In figure to the left, the retriever is initially unable to find a paragraph that directly answers the question, however it finds a paragraph which gives a different name for the disease allowing it to find a more relevant paragraph that directly answers the query. In the figure to the right, after the query reformulation, both the retrieved paragraphs have the correct answer string. Since we aggregate (sum) the scores of spans, this leads to an increase in the score of the right answer span (Demeter) to be the maximum.

## 6 CONCLUSION

This paper introduces a new framework for open-domain question answering in which the retriever and the reader *iteratively* interact with each other. The resulting framework improved performance of machine reading over the base reader model uniformly across four open-domain QA datasets. We also show that our fast retrieval method can scale upto millions of paragraphs, much beyond the current capability of existing open-domain systems with a trained retriever module. Finally, we show that our method is agnostic to the architecture of the machine reading system provided we have *access* to the token level hidden representations of the reader. Our method brings an increase in performance to two popular and widely used neural machine reading architectures, Dr.QA and BiDAF.

## ACKNOWLEDGEMENTS

This work is funded in part by the Center for Data Science and the Center for Intelligent Information Retrieval, and in part by the National Science Foundation under Grant No. IIS-1514053 and in part by the International Business Machines Corporation Cognitive Horizons Network agreement number W1668553 and in part by the Chan Zuckerberg Initiative under the project Scientific Knowledge Base Construction. Any opinions, findings and conclusions or recommendations expressed in this material are those of the authors and do not necessarily reflect those of the sponsor.

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
