# OpenReview forum: "Multi-step Retriever-Reader Interaction for Scalable Open-domain Question Answering"
_ICLR.cc/2019/Conference_

### Official Review · AnonReviewer2 · 2018-10-21
**New framework, but weak comparison**

**Rating:** 7
**Confidence:** 4

**Review:**

This paper introduces a new framework to interactively interact document retriever and reader for open-domain question answering. While retriever-reader framework was often used for open-domain QA, this bi-directional interaction between the retriever and the reader is novel and effective because
1) If the retriever fails to retrieve the right document at the first step, the reader can give a signal to the retriever so that the retriever can recover its mistake at the next step
2) The idea of `reader state` from the reader to the retriever is new
3) The retriever use question-independent representation of paragraphs, which does not require different representation depending on the question and makes the framework easily scalable.

Strengths
1) The idea of multi-step & bi-directional interaction between the retriever and the reader is novel enough (as mentioned above). The paper contains enough literature studies on existing retriever-reader framework in open-domain setting, and clearly demonstrates how their framework is different from them.
2) The authors run the experiments on 4 different dataset, which supports the argument about the framework’s effectiveness.

Weakness
1) The authors seem to highlight multi-step `reasoning`, while it is not `reasoning` in my opinion. Multi-step reasoning refers to the task which you need evidence from different documents, and/or you need to find first evident to find the second evidence from a different document. I don’t think the dataset here are not multi-step reasoning dataset, and the authors seem not to claim it either. Therefore, I recommend using another term (maybe `multi-step interaction`?) instead of `multi-step reasoning`.
2) While the idea of multi-step interaction and how it benefits the overall performance is interesting, the analysis is not enough. Figure 3 in the paper does not have enough description — for example, I got the left example means step 2 recovers the mistake from step 1, but what does the right example mean?

Questions on result comparison
1) On TriviaQA (both open and full), the authors mentioned the result is on hidden test set — did you submit it to the leaderboard? I don’t see the same numbers on the TriviaQA leaderboard. Also, the authors claim they are SOTA on TriviaQA, but there are higher numbers on the leaderboard (which are submitted prior to the ICLR deadline).
2) There are other published papers with higher result on Quasar-T, SearchQA and TriviaQA (such as https://aclanthology.info/papers/P18-1161/p18-1161 and https://arxiv.org/abs/1805.08092) which the authors did not compare with.
3) In Section 4.2, is there a reason for the specific comparison to AQA (5th line), though AQA is not SOTA on SearchQA? I don’t think it means latent space is better than natural language space. They are totally different model and the only intersection is they contains interaction between two submodules.
4) In Section 5, the authors mentioned their framework outperforms previous SOTA by 15% margin on TriviaQA, but what is that? I don’t see 15% margin in Table 2.

Marginal comments:
1) If I understood correctly, `TriviaQA-open` and `TriviaQA-full` in the paper are officially called `TriviaQA-full` and `open-domain TriviaQA`. How about changing the term for readers to better understand the task? Also, in Section 4, the authors said TriviaQA-open is larger than web/wiki setting, but to my knowledge, this setting is part of the wiki setting.
2) It would be great if the authors make the capitalization consistent. e.g. EM, Quasar-T, BiDAF. Also, the authors can use EM instead of `exact match` after they mentioned EM refers to exact match in Section 4.2.

Overall comment
The idea in the paper is interesting, and their model and experiments are concrete. My only worries is that the terms in the paper are confusing and performance comparison are weak. I would like to update the score when the authors update the paper.


Update 11/27/2018
Thanks for the authors for updating the paper. The updated paper have more clear comparisons with other models, with more & stronger experiments with the additional dataset. Also, the model is claimed to perform multi-step interaction rather than multi-step reasoning, which clearly resolves my initial concern. The analysis, especially ablations in varying number of iterations, was helpful to understand how their framework benefits. I believe these make the paper stronger along with its initial novelty in the framework. In this regard, I vote for acceptance.

---

> ### Author Response · Authors · 2018-11-27
> **Response to reviewer 2**
>
> We thank you for your very useful and detailed review. We have significantly updated the writing of the paper to hopefully address all confusion and we’ve also updated the results section of the paper for better comparison. In a nutshell, we have added a section on retriever performance demonstrating the scalability of our approach (sec 5.1). We have improved results for our experiments with BiDAF reader and we have also added new results on the open-domain version of the SQuAD dataset. Below we address your concerns point-by-point.
>
> 1. The authors seem to highlight multi-step `reasoning`, while it is not `reasoning` in my opinion. Multi-step reasoning refers to the task which you need evidence from different documents, and/or you need to find first evident to find the second evidence from a different document. I don’t think the dataset here are not multi-step reasoning dataset, and the authors seem not to claim it either. Therefore, I recommend using another term (maybe `multi-step interaction`?) instead of `multi-step reasoning`.
>
> After much discussion among us, we have arrived to an agreement with your comment. We have renamed the title of the paper to “Multi-step Retriever-Reader Interaction for Scalable Open-domain Question Answering”.
> We believe that our framework that supports retriever-reader interaction would be a starting point to build models for multi-hop reasoning but the current datasets do not explicitly need models with such inductive bias. There has been some very recent efforts in this direction such as HotpotQA -- but this dataset was very recently released (after the ICLR submission deadline).
>
> 2. While the idea of multi-step interaction and how it benefits the overall performance is interesting, the analysis is not enough. Figure 3 in the paper does not have enough description — for example, I got the left example means step 2 recovers the mistake from step 1, but what does the right example mean?
>
> We have significantly updated this section of the paper with much more analysis. We have included a new section on analysis of results (Sec 4.3) in which we quantitatively measure if the iterative interaction between the retriever and the reader is able to retrieve better context for the reader. We have also updated Figure 2 to report the results of our model for steps = {1, 3, 5, 7} for SearchQA, Qusar-T and TriviaQA-unfiltered.
> To answer your specific question about the second example from figure 3, after the query reformulation the new paragraph that was added also has the right answer string, i.e. the total occurrence of the correct answer span increased after the reformulation step. Since we sum up the scores of spans, this led to the overall increase in the score of the right answer span (Demeter, in Figure 3)  to be the maximum. We have explained this in the text of the paper.
>
> 3. On TriviaQA (both open and full), the authors mentioned the result is on hidden test set — did you submit it to the leaderboard? I don’t see the same numbers on the TriviaQA leaderboard. Also, the authors claim they are SOTA on TriviaQA, but there are higher numbers on the leaderboard (which are submitted prior to the ICLR deadline).
>
> We apologize for the confusion about this experiment. Ours and the reported baseline results are on the “TriviaQA-unfiltered” dataset (unfiltered version in http://nlp.cs.washington.edu/triviaqa/), for which there is no official leaderboard. The unfiltered version is built for open-domain QA. The evidence for each question in this setting are top 10 documents returned by Bing search results along with the Wikipedia pages of entities in the question. In the web setting, each question is associated with only one web document and in the Wikipedia setting, each question is associated with the wiki pages of entities in the question (1.78 wiki pages per query on avg.) Thus, the unfiltered setting has much more number of paragraphs than the individual web/wiki setting.  Moreover, there is no guarantee that every document in the evidence will contain the answer making this setting even more challenging. However we did submit our model predictions to the TriviaQA admin who emailed us back the result on the hidden test set and to the best of our knowledge, we achieve the highest result on this setting of TriviaQA. We have updated the paper by naming this experiment TriviaQA-unfiltered and have clarified other details.

---

> > ### Author Response · Authors · 2018-11-27
> > **Response to reviewer 2 (continued)**
> >
> > Response to Reviewer 2 (continued from before)
> > 4. There are other published papers with higher result on Quasar-T, SearchQA and TriviaQA (such as https://aclanthology.info/papers/P18-1161/p18-1161 and https://arxiv.org/abs/1805.08092) which the authors did not compare with.
> >
> > Work by (Min, Zhong, Socher, Ziong, 2018) has results on TriviaQA-wikipedia setting. Our results are on the unfiltered setting of TriviaQA as we mentioned in the previous response, hence the results are not comparable. However, their results on SQuAD-open is comparable to our new experiments on SQuAD and we have added it in Table 2.
> > We also have results of DS-QA (Lin, Ji, Liu, Sun, 2018) in Table 2. They indeed have better results than us on SearchQA and we outperform them in TriviaQA-unfiltered. We tried to reproduce their results on Quasar-T with their code base and shared hyperparameter setting, but we could not reproduce it. However, for fairness, we have reported both their reported scores and our scores in the latest version of the paper.
> >
> > 5. In Section 5.2, is there a reason for the specific comparison to AQA (5th line), though AQA is not SOTA on SearchQA? I don’t think it means latent space is better than natural language space. They are totally different model and the only intersection is they contains interaction between two submodules.
> >
> > Active Question Answering (AQA) propose a model in which an query reformulation agent sits between an user and a black box “QA” system. The agent probes the reader model (BiDAF) with (N=20) reformulations of the initial natural language query and aggregates the returned evidence to yield the best answer. The reformulation is done by a seq2seq model. In our method, the query reformulation is done by a gated recurrent unit to the initial query vector and this update is conditioned on the current state of the reader. By using the same reader architecture (BiDAF) in our experiments, we find significant improvements on SearchQA and other datasets.
> > We have updated the paper to make this distinction very clear. We only wanted to convey that our strategy of query reformulation yield better empirical results than the query reformulation strategy adopted by AQA. However we do agree with you that there is no specific reason to compare this in the experiment section and we have removed it from there and added more relevant results.
> >
> > 6. In Section 5, the authors mentioned their framework outperforms previous SOTA by 15% margin on TriviaQA, but what is that? I don’t see 15% margin in Table 2.
> >
> > This is a miscalculation and was a huge oversight from our part. The relative increase from the previous best result is 9.5% (61.66 - 56.3)/56.3. We mistakenly calculated the improvement from results of R^3 which is a 14.98% (61.66 - 53.7)/53.7 relative increase. We have fixed it.
> >
> > If I understood correctly, `TriviaQA-open` and `TriviaQA-full` in the paper are officially called `TriviaQA-full` and `open-domain TriviaQA`. How about changing the term for readers to better understand the task? Also, in Section 4, the authors said TriviaQA-open is larger than web/wiki setting, but to my knowledge, this setting is part of the wiki setting.
> >
> > Thanks for the suggestion. Yes we agree, the naming convention we chose was confusing.  `TriviaQA-full` is better known as TriviaQA-unfiltered, so we adopted that name. And for the experiment with 1.6M paragraphs per query, we have renamed it to TriviaQA-open, as per your suggestion.
> >
> > It would be great if the authors make the capitalization consistent. e.g. EM, Quasar-T, BiDAF. Also, the authors can use EM instead of `exact match` after they mentioned EM refers to exact match in Section 5.2.
> > We have fixed this, thanks!

---

> > > ### Comment · AnonReviewer2 · 2018-11-27
> > > **Increasing the score**
> > >
> > > Thanks for the authors for updating the paper. The updated paper have more clear comparisons with other models, with more & stronger experiments with the additional dataset. Also, the model is claimed to perform multi-step interaction rather than multi-step reasoning, which clearly resolves my initial concern. The analysis, especially ablations in varying number of iterations, was helpful to understand how their framework benefits. I believe these make the paper stronger along with its initial novelty in the framework. In this regard, I vote for acceptance.

---

> > > > ### Author Response · Authors · 2018-11-27
> > > > **Thank you!**
> > > >
> > > > Thank you for your insightful comments which helped make the paper a lot better.

---

### Official Review · AnonReviewer1 · 2018-11-02
**Very interesting idea; needs more details and better evaluation**

**Rating:** 6
**Confidence:** 4

**Review:**

The authors improve a retriever-reader architecture for open-domain QA by iteratively retrieving passages and tuning the retriever with reinforcement learning. They first learn vector representations of both the question and context, and then iteratively change the vector representation of the question to improve results. I think this is a very interesting idea and the paper is generally well written.

I find some of the description of the models, methods and training is lacking detail. For example, their should be more detail on how REINFORCE was implemented; e.g. was a baseline used?

I am not sure about the claim that their method is agnostic to the choice of machine reader, given that the model needs access to internal states of the reader and their limited results on BiDAF.

The presentation of the results left a few open questions for me:

  - It is not clear to me which retrieval method was used for each of the baselines in Table 2.
  - Why does Table 2 not contain the numbers obtained by the DrQA model (both using the retrieval method from the DrQA method and their method without reinforcement learning)? That would make their improvements clear.
  - Moreover, for TriviaQA their results and the cited baselines seem to all perform well below to current top models for the task (cf. https://competitions.codalab.org/competitions/17208#results).
  - I would also like to see a better analysis of how the number of steps helped increase F1 for different models and datasets. The presentation should include a table with number of steps and F1 for different step numbers they tried. (Figure 2 is lacking here.)
  - In the text, the authors claim that their result shows that natural language is inferior to 'rich embedding spaces'. They base this on a comparison with the AQA model. There are two problems with this claim: 1) The two approaches 'reformulate' for different purposes, retrieval and machine reading, so they are not directly comparable. 2) Both approaches use a 'black box' machine reading model, but the authors use DrQA as the base model while AQA uses BiDAF. Indeed, since the authors have an implementation of their model that uses BiDAF, an additional comparison based on matched machine reading models would be interesting.
- Generally, it would be great to see more detailed results for their BiDAF-based model as well.

---

> ### Author Response · Authors · 2018-11-27
> **Response to reviewer 1**
>
> We sincerely thank you for your insightful comments and we’re glad that you found our approach interesting. Based on your comments, we have significantly improved the writing of the paper with more details and have added more evaluation. Below we address your concerns point-by-point.
>
> - I find some of the description of the models, methods and training is lacking detail. For example, their should be more detail on how REINFORCE was implemented; e.g. was a baseline used?
>
> We have significantly updated the model section of our paper to include more details about methods and training (Sec 2 & 3). To answer your specific question about use of variance reduction baseline with REINFORCE -- In question answering settings, it has been noted by previous work such as  Shen et al., (2017) that common variance reduction techniques don’t work well. We also tried experimenting with a commonly used baseline - the average reward in a mini-batch, but found that it significantly degrades the final performance.
>
> I am not sure about the claim that their method is agnostic to the choice of machine reader, given that the model needs access to internal states of the reader and their limited results on BiDAF.
>
> We agree with you and based on your comments we have made this absolutely clear in the paper. Our method needs access to the internal token level representation of the reader model in order to construct the current state. The current API of machine reading models only return the span boundaries of the answer, but for our method, it needs to return the internal state as well. What we wanted to convey is, our model does not depend/need any neural architecture re-designing to an existing reader model. To show the same, we experimented and showed improvements with two popular and widely used reader architectures - DrQA and BiDAF.
> Regarding results of BiDAF -- During submission we ran out of time and hence we could not tune the BiDAF model. But now the results of BiDAF have improved a lot and as can be seen from (Table 2, row 9), the results of BiDAF are comparable to that of DrQA.
>
> It is not clear to me which retrieval method was used for each of the baselines in Table 2.
>
> We report the best performance for each of our baseline that is publicly available. Most of the results for the baseline (except DS-QA) are taken as reported in the R^3 paper. We briefly describe the retrieval method used by the baselines below:
> (a) R^3 and DS-QA, like us, has a trained retriever module. R^3 retriever is based on the Match-LSTM model and DS-QA is based on DrQA model (more details in the respective papers). However, their retrievers compute query dependent para representation and hence don’t scale as we experimentally demonstrate in Fig 2.
> (b) AQA, GA and BiDAF lack an explicit retriever module. They concatenate all paragraphs in the context and feed it to their respective machine reading module. Since the reader has to find the answer from possible very large context (because of concatenation), these models have lower performance as can be seen from Table 2.
>
> Why does Table 2 not contain the numbers obtained by the DrQA model (both using the retrieval method from the DrQA method and their method without reinforcement learning)? That would make their improvements clear.
>
> Thanks for suggesting this experiment! We ran the experiment and results are in (Table 2, row 7). We trained a DrQA baseline model and the results indeed suggest that multi-step reasoning give uniform boost in performance across all datasets.

---

> > ### Author Response · Authors · 2018-11-27
> > **Response to Reviewer 1 (continued)**
> >
> > Response to Reviewer 1 (continued from before)
> >
> > Moreover, for TriviaQA their results and the cited baselines seem to all perform well below to current top models for the task (cf. https://competitions.codalab.org/competitions/17208#results).
> >
> > We apologize for the confusion about this experiment. Ours and the reported baseline results are on the “TriviaQA-unfiltered” dataset (unfiltered version in http://nlp.cs.washington.edu/triviaqa/), for which there is no official leaderboard. The unfiltered version is built for open-domain QA. The evidence for each question in this setting are top 10 documents returned by Bing search results along with the Wikipedia pages of entities in the question. In the web setting, each question is associated with only one web document and in the Wikipedia setting, each question is associated with the wiki pages of entities in the question (1.78 wiki pages per query on avg.) Thus, the unfiltered setting has much more number of paragraphs than the individual web/wiki setting.  Moreover, there is no guarantee that every document in the evidence will contain the answer making this setting even more challenging. However, we did submit our model predictions to the TriviaQA admin who emailed us back the result on the hidden test set. We have updated the paper by naming this experiment TriviaQA-unfiltered and have clarified other details.
> >
> > I would also like to see a better analysis of how the number of steps helped increase F1 for different models and datasets. The presentation should include a table with number of steps and F1 for different step numbers they tried. (Figure 2 is lacking here.)
> >
> > We have included a detailed result in figure 2 where we note the results of our model for steps = {1, 3, 5, 7} for SearchQA, Qusar-T and TriviaQA-unfiltered. The key takeaway from the result is that multi-step interaction uniformly increases the performance across all the datasets.
> >
> > In the text, the authors claim that their result shows that natural language is inferior to 'rich embedding spaces'. They base this on a comparison with the AQA model. There are two problems with this claim: 1) The two approaches 'reformulate' for different purposes, retrieval and machine reading, so they are not directly comparable. 2) Both approaches use a 'black box' machine reading model, but the authors use DrQA as the base model while AQA uses BiDAF. Indeed, since the authors have an implementation of their model that uses BiDAF, an additional comparison based on matched machine reading models would be interesting.
> >
> > We have now reported the results of our method with a BiDAF reader on SearchQA (row 9, table 2) and have shown that our method outperforms AQA by a significant margin when both the model uses the same reader architecture (BiDAF).
> >
> > Active Question Answering (AQA) propose a model in which an query reformulation agent sits between an user and a black box “QA” system. The agent probes the reader model (BiDAF) with (N=20) reformulations of the initial natural language query and aggregates the returned evidence to yield the best answer. The reformulation module is trained end to end using policy gradients to maximize the F1 of the reader. In our method as well, the query reformulation is done to the initial query vector to maximize the F1 of the reader. In other words, both methods are reformulating to improve retrieval. By using the same reader architecture (BiDAF) in our experiments, we find significant improvements on SearchQA. We have updated the paper to make this distinction very clear.

---

### Official Review · AnonReviewer3 · 2018-11-02
**Interesting and encouraging results but limited novelties**

**Rating:** 6
**Confidence:** 5

**Review:**

The paper proposes a multi-document extractive machine reading model and algorithm. The model is composed of 3 distinct parts. First, the document retriever and the document reader that are states of the art modules. Then, the paper proposes to use a "multi-step-reasoner" which learns to reformulate the question into its latent space wrt its current value and the "state" of the machine reader.

In the general sense, the architecture can be seen as a specific case of a memory network. Indeed, the multi-reasoner step can be seen as the controller update step of a memory network type of inference. The retriever is the attention module and the reader as the final step between the controller state and the answer prediction.

The authors claim the method is generic, however, the footnote in section 2.3 mentioned explicitly that the so-called state of the reader assumes the presence of a multi-rnn passage encoding. Furthermore, this section 2.3 gives very little detailed about the "reinforcement learning" algorithms used to train the reasoning module.

Finally, the experimental section, while giving encouraging results on several datasets could also have been used on QAngoroo dataset to assess the multi-hop capabilities of the approach. Furthermore, very little details are provided regarding the reformulation mechanism and its possible interpretability.

---

> ### Author Response · Authors · 2018-11-27
> **Response to reviewer 3**
>
> We thank you for your helpful reviews. We have significantly updated the writing of the paper to hopefully address all confusion and we’ve also updated the results section of the paper for better comparison. In a nutshell, we have added a section on retriever performance demonstrating the scalability of our approach (sec 4.1). We have improved results for our experiments with BiDAF reader and we have also added new results on the open-domain version of the SQuAD dataset.
>
> > In the general sense, the architecture can be seen as a specific case of a memory network.  Indeed, the multi-reasoner step can be seen as the controller update step of a memory network type of inference. The retriever is the attention module and the reader as the final step between the controller state and the answer prediction.
>
> We agree with you and think its a valid way of viewing our framework. We have updated and cited memory networks in our paper (Sec 4) . However, we would like to point out that most memory network architectures are based on soft-attention, but in our case the retriever actually makes a “hard selection” of the top-k paragraphs and hence for the same reason, we have to train it via reinforcement learning.
>
> > The authors claim the method is generic, however, the footnote in section 2.3 mentioned explicitly that the so-called state of the reader assumes the presence of a multi-rnn passage encoding. Furthermore, this section 2.3 gives very little detailed about the "reinforcement learning" algorithms used to train the reasoning module.
>
> We agree with you and based on your comments we have made this absolutely clear in the paper. Our method needs access to the internal token level representation of the reader model in order to construct the current state. The current API of machine reading models only return the span boundaries of the answer, but for our method, it needs to return the internal state as well. What we wanted to convey is, our model does not depend/need any neural architecture re-designing to an existing reader model. To show the same, we experimented and showed improvements with two popular and widely used reader architectures - DrQA and BiDAF.
> Regarding results of BiDAF -- During submission we ran out of time and hence we could not tune the BiDAF model. But now the results of BiDAF have improved a lot and as can be seen from (Table 2, row 9), the results of BiDAF are comparable to that of DrQA.
> We have also significantly updated the model section of our paper to include more details about methods and training (Sec 2 & 3) with details about our policy gradient methods and training procedure.
>
> > Finally, the experimental section, while giving encouraging results on several datasets could also have been used on QAngaroo dataset to assess the multi-hop capabilities of the approach.
>
> We did not consider QAngaroo for the following reasons -- (a) The question in QAngaroo are based on knowledge base relations and are not natural language questions. This makes the dataset a little synthetic in nature and we were unsure if our query reformulation strategy would work in this synthetic setting. (b) In this paper, we have tried to focus on datasets for open domain settings where the number of paragraphs per query is large (upto millions). QAngaroo on the other hand is quite small in that respect (avg of 13.7 paragraphs per question). We were unsure, that in this small setting, if we would see significant gains by doing query reformulation.
>
> We have shown the effectiveness of our model in 4 large scale datasets including new results on SQuAD-open since submission. We sincerely hope, we will not be penalized for not showing the effectiveness of our model on enough number of datasets.
>
> > Furthermore, very little details are provided regarding the reformulation mechanism and its possible interpretability.
>
> We have significantly updated this section of the paper. We have added a whole new section (Sec 5.3) with detailed analysis of the effect of query reformulation. In Table 4, we quantitatively measure if the iterative interaction between the retriever and reader is able to retrieve better context for the reader.

---

### Author Response · Authors · 2018-11-27
**Summary of updates**

Based on the insightful feedback from our reviewers, we’ve updated our paper. Below we summarize the general changes.

Writing and analysis of results: We have significantly improved the writing of our paper, especially the model (Sec 2, Sec 3) and the experiments section (Sec 5). We have added the details of our training methodology (e.g. details of reinforcement learning and various hyperparameters). In the experiments section, we have included a new section on analysis of results (Sec 5.3) in which we quantitatively measure if the iterative interaction between the retriever and reader is able to retrieve better context for the reader (Table 4)

Performance of paragraph retriever: We have added a new section on the performance of the paragraph retriever (Sec 4.1). We show that our retriever architecture based on fast nearest neighbor search can scale to corpus containing millions of paragraphs where as retrievers of current best-performing models cannot scale to that size.

New BiDAF results: During initial submission we ran out of time and could not tune our implementation of the BiDAF model. But since, the results of BiDAF have improved a lot and are comparable to that of DrQA (Table 2).

New results on SQuAD-open: We have also added new results on another popular dataset -- the open domain setting of SQuAD. Following the setting of Chen et al., (2017), we were able to demonstrate that our framework of multi-step-interaction improves the exact match performance of a base DrQA model from 27.1 to 31.9.

Change in title:. Following the comment by reviewer 2, we have renamed the title of the paper to “Multi-step Retriever-Reader Interaction for Scalable Open-domain Question Answering”.
We believe that our framework that supports retriever-reader interaction would be a starting point to build models for multi-hop “reasoning” but the current datasets do not explicitly need models with such inductive bias. Hence it will be more appropriate for our work to have this title.

---

### Public Comment · (anonymous) · 2018-12-10
**Model implementation and source code**

This paper is very interesting and we're are doing follow-up research. Could the authors update their link to their source code?  The current link doesn't seem to work. Thanks a lot!

---

> ### Author Response · Authors · 2018-12-10
> **Re:**
>
> Thanks for your comment!. Right now the link is intentionally anonymized. We will release the code once the decision on the paper is finalized. Thank you for your interest!

---

> > ### Public Comment · (anonymous) · 2018-12-10
> > **Thank you!**
> >
> > That would be really helpful! Thanks for your update!

---

> > > ### Author Response · Authors · 2019-03-14
> > > **Re:**
> > >
> > > I apologize for the delay in releasing the code. The code and pretrained models are available here (https://github.com/rajarshd/Multi-Step-Reasoning).
> > >
> > > Thanks!
> > > Rajarshi

---

> > ### Public Comment · (anonymous) · 2018-12-21
> > **Congratulations!**
> >
> > Congratulations on your acceptance! I thought it was work done by Mr.Das and Prof.McCallum(and it really is). Your writing style and research idea are quite consistent with MINERVA. I wish you could open source soon so that we can learn from it like MINERVA, and catch deadline of ACL (as well as NAACL rebuttal).

---

> > > ### Author Response · Authors · 2018-12-22
> > > **Thank you!**
> > >
> > > Thanks again!. This work was very much a joint effort with Shehzaad Dhuliawala and Manzil Zaheer. We are planning on open-sourcing the code as soon as possible. The holidays might delay it by a week but if you need it sooner, feel free to email us and we will work with you.

---

### Public Comment · (anonymous) · 2019-01-25
**Opensourcing?**

Hi, any update on the source code?

---

> ### Public Comment · (anonymous) · 2019-02-12
> **Open Source model/code**
>
> Agreed will be great if you can open source the code and models soon!

---

> > ### Author Response · Authors · 2019-03-14
> > **Sorry for the delay**
> >
> > Due to personal deadlines, releasing the code got delayed, but I have opensourced the code and pretrained models here -- https://github.com/rajarshd/Multi-Step-Reasoning
> >
> > Thanks!,
> >
> > Rajarshi

---

### Meta-Review · Area_Chair1 · 2018-12-13
**An interesting approach to open domain QA using query rewriting in latent space**

**Confidence:** 3
**Recommendation:** Accept (Poster)

**Metareview:**


pros:
- novel idea for multi-step QA which rewrites the query in embedding space
- good comparison with related work
- reasonable evaluation and improved results

cons:

There were concerns about missing training details, insufficient evaluation, and presentation.  These have been largely addressed in revision and I am recommending acceptance.